# How do smoking cessation medicines compare with respect to their neuropsychiatric safety? A protocol for a systematic review, network meta-analysis and cost-effectiveness analysis

Kyla H Thomas,[1] Deborah Caldwell,[1] Michael N Dalili,[1] David Gunnell,[1] Marcus R Munafò,[2,3] Matt Stevenson,[4] Nicky J Welton[1]

► Prepublication history and additional material are available. To view these files, please visit the journal online (http://dx.doi.org/10.1136/bmjopen-2016-015414).

[1]School of Social and Community Medicine, University of Bristol, Bristol, UK
[2]MRC Integrative Epidemiology Unit, University of Bristol, Bristol, UK
[3]UK Centre for Tobacco and Alcohol Studies, School of Experimental Psychology, University of Bristol, Bristol, UK
[4]School of Health and Related Research, University of Sheffield, Sheffield, UK

**Correspondence to**
Kyla H Thomas; kyla.thomas@bristol.ac.uk

## ABSTRACT

**Introduction** Cigarette smoking is one of the leading causes of early death in the UK and worldwide. Public health guidance recommends the use of varenicline, bupropion and nicotine replacement therapy (NRT) as smoking cessation aids in the UK. Additionally, the first electronic cigarette has been licensed for use as a smoking cessation medicine. However, there are ongoing concerns about the safety of these medicines. We present a protocol for a systematic review and network meta-analysis (NMA) to determine how these smoking cessation medicines compare to each other with respect to their neuropsychiatric safety in adult smokers. Secondary aims include updating the evidence regarding the effectiveness and cardiovascular safety of these medicines for use in a cost-effectiveness analysis.

**Methods and analysis** We will include randomised controlled trials and observational studies with control groups comparing monotherapy with varenicline, bupropion, NRT or electronic cigarette and combination therapies to each other, placebo or usual care. The primary composite safety outcome will be serious adverse events, defined as events that resulted in death, were life threatening, required hospitalisation or resulted in significant disability or congenital/birth defect. The preferred effectiveness outcome will be sustained smoking cessation defined as abstinence for a minimum of 6 months as determined by biochemical validation. We will include trials identified by previous reviews and search relevant databases for newly published trials as well as contacting study authors to identify unpublished information. We will conduct fixed-effect and random-effect meta-analyses for each pairwise comparison of treatments and outcome; where these estimates differ, we will consider reasons for heterogeneity, quantified using the between-study variance ($\tau^2$). For each outcome, we will construct a NMA in a Bayesian framework which will be compared with the pair-wise results, allowing us to rank treatments. The effectiveness estimates from the NMA will be entered into a probabilistic economic model.

**Ethics and dissemination** Ethics approval is not required for this evidence synthesis study as it involves analysis of secondary data from randomised controlled trials and observational studies. The review will make an

### Strengths and limitations of this study

► This study will be the first comprehensive review of the neuropsychiatric safety of smoking cessation medicines in relation to each other and will include randomised controlled trials of any duration as well as observational studies with control groups. Including observational studies will allow us to identify sufficient data for the network meta-analysis and improve the precision of estimates of adverse events.

► This study will include trials of combined therapies of smoking cessation medicines, not included in previous network meta-analyses, as the analysis of both safety and efficacy data on coprescribing could be of important to regulators.

► This study will produce updated cost-effectiveness analyses to estimate which smoking cessation medicine is the most cost-effective in UK settings.

► We will use the Cochrane tool for assessing the risk of bias and consider using the risk of bias in non-randomised studies of interventions tool to assess the risk of bias in randomised controlled trials and observational studies, respectively.

► This study will be limited to products licensed for use as smoking cessation aids in the UK.

important contribution to the knowledge base around the effectiveness, safety and cost-effectiveness of smoking cessation medicines. Results will be disseminated to the general public, healthcare practitioners and clinicians, academics, industry and policy makers.
**PROSPERO registration number** CRD42016041302.

## INTRODUCTION

Cigarette smoking is one of the leading causes of early death in the UK and worldwide.[1 2] Each year, more than 100 000 people will die in the UK from smoking related diseases.[3] The National Institute for Health and Care Excellence (NICE) public health guidance

(published in 2008) recommends the use of three medicines, varenicline, bupropion and nicotine replacement therapy (NRT), as aids to quitting smoking in the UK.[4] Additionally, in the late 2015, the Medicines and Healthcare Products Regulatory Agency (MHRA) approved the use of British American Tobacco's electronic cigarette 'e-Voke' as a smoking cessation medicine.[5] Since the publication of the original NICE guidance, there have been ongoing concerns about the safety of the smoking cessation medicines, with particular respect to the neuropsychiatric safety of varenicline. Severe safety warnings (black triangle and black box warnings) regarding a potential increased risk of serious neuropsychiatric adverse events (depression, suicidal ideation and suicidal behaviour) in patients prescribed these medicines have previously been issued by regulatory agencies such as the MHRA in the UK and the Food and Drug Administration (FDA) in the USA.[6 7] These safety warnings were based on spontaneous reports to the UK Yellow Card Scheme and the FDA Adverse Events Reporting Database. Previous research into the neuropsychiatric safety of these medicines has provided inconsistent findings, adding to the debate.[8] In April 2016, the results of Evaluating Adverse Events in a Global Smoking Cessation Study (EAGLES) trial,[9] a randomised controlled trial (RCT) that randomised 8144 smokers to receive varenicline, transdermal NRT patch, bupropion or placebo, were published. The trial's findings provided strong evidence that both varenicline and bupropion do not cause an increase in neuropsychiatric adverse events relative to nicotine patch or placebo. Subsequently, the European Medicines Agency lifted the warning about possible suicidal risks from varenicline in April 2016,[10] followed by the FDA's decision to remove the black box warnings on varenicline's labelling in December 2016. While the EAGLES trial may be the largest global clinical trial of smoking cessation medicines, its sample size is limited relative to much larger observational cohort studies.[11 12] Therefore, synthesising these findings with those of other RCTs, as well as synthesising the results of observational studies, will offer a more comprehensive review of the neuropsychiatric safety of smoking cessation medicines. With the MHRA's licensing of the first electronic cigarette as a smoking cessation medicine, and given the popularity of electronic cigarettes with an estimated 2.8 million adult users in Great Britain,[13] it is important to review their effectiveness as a smoking cessation aid as well as their safety.

The health benefits of smoking cessation have been well documented. Varenicline has been shown to be the most clinically effective monotherapy for long-term smoking abstinence (>6 months).[14] However, the number of prescription items of varenicline dispensed in England decreased by 25% from a peak of approximately one million prescriptions in 2011 to almost 742 000 prescriptions in 2013,[15] possibly reflecting ongoing fears among prescribers and patients regarding varenicline's neuropsychiatric safety. It is important for patients, prescribers

and regulators to know how smoking cessation medicines compare with each other, with particular respect to their neuropsychiatric safety, to enable smokers wanting to quit and their healthcare professionals to make informed decisions about the risks and benefits of the different pharmacological treatments. To date, there has been no comprehensive analysis of the neuropsychiatric safety of the smoking cessation medicines; previous systematic reviews have focused on comparisons of varenicline with placebo.[16–18] In addition, the cost-effectiveness of these medicines in UK settings has not been investigated using the most up-to-date evidence. These analyses are important to inform the overall risk–benefit evaluation of the different smoking cessation medicines.

The ongoing debate regarding the neuropsychiatric safety of drugs for smoking cessation among drug regulators, researchers, prescribers and patients may be due to the inconsistent research findings in this area.[8] Whereas studies without control groups (such as those using adverse event reporting data and case studies)[19–21] have reported increased risks of self-reported depression and suicidal or self-injurious behaviour in patients prescribed varenicline and bupropion, studies with control groups (such as observational cohort studies and experimental study designs, mainly RCTs and systematic reviews of RCTs) have reported the opposite, and found no evidence of an increased risk of these severe neuropsychiatric outcomes in patients prescribed these medicines.[11 16–18 22–24] However, there are important limitations associated with each of these study designs. First, studies which use spontaneous adverse event reports are limited by several factors. These include the severity of the adverse event (severe adverse events are more likely to be reported than less serious events),[25] the length of time that the drug has been available (adverse events with newer drugs are more likely to be reported than events occurring with older drugs for the same indication) and media publicity about a drug (media reports often lead to increased reporting of adverse events to the Yellow Card Scheme, known as stimulated reporting).[26 27] Second, observational cohort studies are prone to the effects of confounding by indication, which raises concerns about the validity of their findings.[28] Confounding by indication may occur if an observed association between smoking cessation medicines and serious psychiatric events such as suicide is explained if the patient case mix differs among those taking different smoking cessation medications.[11 12 29] In two of the earlier cohort studies, varenicline and bupropion were found to be associated with a decreased risk of death from all causes compared with NRT[11 22]; this reported protective effect was most likely due to residual confounding in the studies.

Experimental studies are less likely to suffer from uncontrolled confounding; however, to date, systematic reviews and meta-analyses of RCTs have mainly focused on comparing the neuropsychiatric safety of varenicline monotherapy with placebo.[16–18 30] Although this is an important research question, patients are unlikely to

be prescribed placebo in real-life settings to help them quit smoking. Therefore, the neuropsychiatric safety of varenicline compared with other smoking cessation drugs is likely to be of greater relevance to patients, prescribers and regulators. To date, there have been no comprehensive reviews of the neuropsychiatric safety of the smoking cessation medicines in relation to each other. In addition, there have been no recent cost-effectiveness analyses that have fully accounted for adverse events in order to determine which UK-licensed smoking cessation medicine is estimated to be the most cost-effective in UK settings. The aim of this study is to determine how smoking cessation medicines compare with each other with respect to their neuropsychiatric safety in adult smokers. Secondary aims include updating the evidence regarding the effectiveness and cardiovascular safety of these medicines for use in a cost-effectiveness analysis.

## METHODS AND ANALYSIS
### Criteria for screening studies
#### Study designs
For the safety network meta-analyses, RCTs of any duration will be included in addition to observational studies with control groups. Uncontrolled observational studies (eg, case reports and case series) will be excluded.

For the effectiveness network meta-analyses, we will include RCTs of greater than 6 months' duration in any setting, including, but not limited to, primary care practices, hospitals including inpatient and outpatient clinics, universities, workplace clinics, nursing or residential homes. Multiarm trials will be included while crossover trials, non-randomised trials, quasi-randomised trials and interrupted time series analyses will be excluded from the effectiveness analyses.

#### Participants
We will include smokers aged 18 years and over of all ethnicities who are seeking to quit smoking using UK-licensed smoking cessation therapies. This includes adult smokers accessing local authority stop smoking services. We will also include smokeless tobacco users. We will exclude studies with participants less than 18 years of age as varenicline, bupropion and electronic cigarettes are only licensed for use in adults in the UK. Non-smoking populations will be excluded as well as pregnant and breastfeeding women, as varenicline and bupropion are not licensed for use in these groups in the UK. If sufficient evidence is identified, we plan to consider the following subgroups in all analyses: those with psychiatric illness (for example, depression, schizophrenia, bipolar disorder, substance misuse), cardiovascular disease (for example, peripheral vascular disease, acute coronary syndromes and postmyocardial infarction), chronic obstructive pulmonary disease, diabetics, heavy smokers (defined as people who smoke >20 cigarettes per day) and those with previous quit attempts.

#### Interventions
The smoking cessation medicines that we will assess include varenicline, bupropion, nicotine replacement therapy and electronic cigarette as monotherapy and in combination treatment (eg, varenicline combined with NRT, varenicline combined with bupropion and bupropion compared with NRT). For NRT, combinations of different formulations given concurrently, for example, patch and gum, will also be assessed. Different dosages of treatments will also be examined (see online supplementary appendix 1 which summarise the main pharmacologic monotherapies for smoking cessation by formulation and dosage using the British National Formulary September 2016 version (http://www.evidence.nhs.uk/formulary/bnf/current) and the MHRA public assessment report for the 'e-Voke').[5] Combination therapies and combinations of NRT formulations will also be included in the analyses.

We will exclude trials of alternative and complementary therapies (eg, hypnotherapy, acupuncture, aromatherapy and herbal therapies) and psychotherapies (unless they are included as cotreatment with a pharmacological intervention).

#### Outcomes
##### Safety analysis
The primary composite safety outcome will be serious adverse events, defined as events that resulted in death, were life threatening, required hospitalisation or resulted in significant disability or congenital/birth defect.[31] Safety outcomes will be grouped under the following headings:

► Neuropsychiatric outcomes. The primary neuropsychiatric outcomes will include: completed suicide, attempted suicide, suicidal ideation, depression and seizures. Secondary neuropsychiatric outcomes will include: abnormal dreams, aggression, anxiety, fatigue, insomnia, irritability, sleep disorders and somnolence.

► Cardiovascular outcomes. The primary cardiovascular outcomes will include: cardiovascular death, non-fatal myocardial infarction (ie, unstable angina) and non-fatal stroke based on the FDA definition.[32] Secondary cardiovascular outcomes will include: transient ischaemic attack, congestive heart failure, palpitations, arrhythmias and thromboembolism (deep vein thrombosis and pulmonary embolism).

Other outcomes will include adverse events such as nausea, headache, dry mouth, skin rash and pruritus.

##### Effectiveness analysis
The primary effectiveness outcome will be sustained smoking cessation, defined as abstinence for a minimum of 6 months as determined by biochemically validated continuous or prolonged abstinence at the longest reported time point in intention to treat analyses, as this is the strictest definition of abstinence. As we are using data from previous Cochrane reviews of smoking cessation medicines,[33–36] we have also chosen this outcome to be consistent with the definitions used

in those reviews. Where these data are not available, we will accept point prevalence abstinence and self-report quit data. We will treat participants who drop out or are lost to follow-up as continuing smokers. We will extract information on abstinence at each time point for which it is reported to allow a survival model to be estimated following the approach used by Chen *et al*[37] and Madan *et al*.[38] Data on reductions in smoking, rather than abstinence, will not be included. Secondary outcomes will include reduction in craving and reduction in withdrawal symptoms.

### Search strategy

We will search the following databases: MEDLINE, EMBASE, PsycINFO, Web of Science, Clinicaltrials.gov and the Cochrane Databases including the Cochrane Database of Systematic Reviews, the Database of Abstracts and Reviews of Effectiveness (updated until March 2015), the Cochrane Central Register of Controlled Trials, the NHS Economic Evaluation Database and the Health Technology Assessment Database. Searches will be conducted with the help of an information specialist and will not include any language restrictions. Non-English language articles will be reviewed by native speakers prior to obtaining a full translation. We will also manually search the reference lists of relevant research articles and previous reviews and communicate with authors in an attempt to identify unpublished information; in a previous study, we had a 75% positive response rate from corresponding authors for studies published after 2006.[17] We will also review the literature to identify disutilities and costs associated with neuropsychiatric and cardiovascular treatment-related adverse events. Acknowledging that we may not find many studies in a smoking cessation population, we will also search for studies reporting disutilities and costs for the same events in other populations.

To identify studies for the safety network meta-analyses, we will build on the basic search strategy included in the cardiovascular network meta-analysis by Mills *et al*[39] (see online supplementary appendix 2 for our full electronic search strategy for observational studies of varenicline in Medline). To identify studies for the effectiveness network meta-analyses, the search strategies from four Cochrane reviews[33–36] (or updated versions where available) will be modified to identify more recent trials for inclusion in the current study in addition to their previously identified trials. Searches for observational studies will not be date limited. We will adapt the Sheffield economic model[30] to incorporate disutilities associated with neuropsychiatric and cardiovascular treatment-related adverse events and update model inputs. The searches used for the network meta-analyses will be rerun with a cost-effectiveness filter to identify studies reporting information on utilities, disutilities, resource use and costs, which will be used to inform the economic evaluation.

### Study selection and data extraction

Search results will be uploaded to Covidence,[40] which we will use to screen abstracts and full text and to resolve disagreements. Reviewers (MND—all papers, KHT and DC—papers equally shared) will independently screen abstracts to determine whether full-text reports should be obtained. The same reviewers will independently identify eligible full-text reports for inclusion. Discrepancies will be resolved by reaching consensus among reviewers. We will include a PRISMA diagram[41] to set out the results of the searches and to indicate the number of included and excluded trials. The reasons for excluding studies following full-text screening will be documented.

Data for studies included following the updated search will be extracted by one reviewer (MND) and checked by coreviewers (KHT and DC). Information will be collected on study design (duration of treatment, description of allocation concealment and blinding), study participants (inclusion and exclusion criteria, country, region and population studied), baseline characteristics (eg, ethnicity, sex and smoking history), intervention and comparison groups (including the smoking cessation intervention, whether or not there was cotreatment, dosage and formulation), our predefined primary and secondary outcomes of interest including measures of effectiveness and safety outcomes, losses to follow-up and study sponsor. All of the extracted information will be summarised in tables. In the event of missing data, we will contact authors by email to ask for original data. Authors of all newly identified studies will be contacted to verify the accuracy of the extracted data.

### Risk of bias assessment

For RCTs, the Cochrane tool for assessing the risk of bias[42] will be used to determine whether there is high, low or unclear risk of bias in the following domains: random sequence generation, allocation concealment, blinding of participants and personnel, blinding of outcome assessment; incomplete outcome data, selective outcome reporting and other sources of bias. For observational studies, we will consider using the risk of bias in non-randomised studies of interventions tool[43] to determine whether there is low, moderate, serious or critical risk of bias or no information in the following domains: confounding, participant selection, intervention classification, intervention deviations, missing data, outcome measurement and selection of the reported result. Reviewers (MND—all papers, KHT and DC—papers equally shared) will independently assess the risk of bias in each of the trials. Discrepancies will be resolved by referring to the original publication and reaching consensus among reviewers. Study authors will be contacted to obtain study protocols and additional information that may not have been published to aid with assessment of the risk of bias.

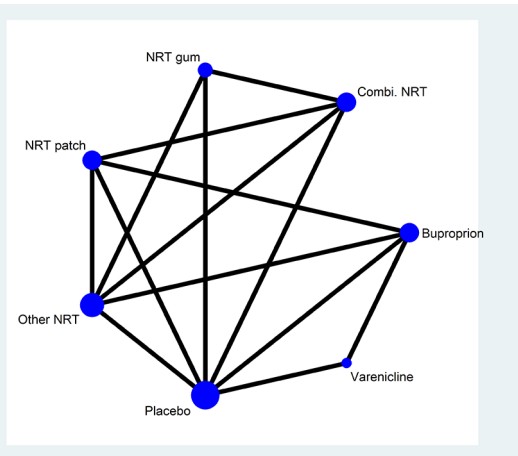

**Figure 1** Network diagram for effectiveness of smoking cessation treatments. Adapted from Cahill *et al*.[14] NRT, nicotine replacement therapy.

## Data synthesis

Effectiveness outcomes will be summarised as risk ratios. Because safety outcomes are rare, we will follow recommendations of Bradburn *et al*[44] and use Peto ORs to compare treatment groups although we will also report risk differences. All results will be reported including 95% CIs. The pairwise and network meta-analyses will be conducted in a Bayesian framework using OpenBUGS software, using code developed by Dias *et al*.[45] We will report results of both fixed-effect and random-effect meta-analyses; between-study heterogeneity will be quantified using the between-study SD ($\tau$). For each separate outcome, we will also construct a network meta-analysis (NMA) that will be compared with the pairwise results. For the safety outcomes, separate analyses will be conducted for RCTs and observational studies. NMA is an extension of standard meta-analysis that allows evidence synthesis to be performed when there are multiple competing interventions available by simultaneously combining evidence from *all* trials reporting that outcome, so long as treatments form a connected network.[46] Figure 1 shows an example of a connected network of seven smoking cessation treatments adapted from a Cochrane overview of reviews of smoking cessation at 6 months.[14] For the NMA, combination therapies will be considered separate interventions but we will explore models for the effects of the component therapies using a main-effect model and a two-way interaction model (allowing pairs of therapies to have either a bigger or smaller effect than would be expected from the sum of their effects alone). For the random-effect NMA, we will assume homogeneous between-study variability across studies. We will assess the goodness of fit of each model to the data by calculating the posterior mean residual deviance. This is defined as the difference between the deviance for the fitted model and the saturated model, where the deviance measures the fit of the model using the likelihood function. The Deviance Information Criterion (DIC), which is equal to the sum of the posterior mean of the residual deviance

and the effective number of parameters pD, will be used as a basis for model comparison.[47] The DIC penalises the posterior mean residual deviance (a measure of model fit) by the effective number of parameters in the model (as measure of complexity) and can therefore be viewed as a trade-off between the fit and complexity of the model.

For the pairwise analyses only, and where the number of studies allow, we will produce funnel plots.

### Assessment of the consistency assumption

Validity of a NMA depends on the assumption that there is no effect modification of the pairwise intervention effects or, that the prevalence of effect modifiers is similar in the different studies. This key assumption has been referred to variously as transitivity,[48] similarity[49] and consistency.[50] We will compile a table of important trial and patient characteristics and visually inspect the 'similarity' of factors we consider likely to modify treatment effect.

We will visually inspect the network diagram to identify the extent of potential inconsistency (the number of loops) and use model fit and selection statistics to informally assess whether it is evident. If inconsistency is suspected, we will explore it formally using a 'node-splitting' approach.[50]

### Health economic modelling

A previously built model[30] will be used to evaluate cost-effectiveness using probabilistic methods and reporting fully incremental analyses that meet NICE reference case.[51] This model has nine health states including those most commonly related to smoking. Where data allow the model will be adapted to allow more than one attempt at quitting and the addition of serious adverse neuropsychiatric and cardiovascular events.

Subgroup analyses will be presented as appropriate. Sensitivity analyses will be performed including and excluding smokeless tobacco users. We will use the Grading of Recommendations Assessment, Development and Evaluation approach to assess the quality of evidence.[52] Evidence for both effectiveness and safety outcomes will be judged for the domains of risk of bias, consistency, precision, reporting bias and directness. Evidence will be ranked as being of high, moderate, low or very low quality.

## CONCLUSION

The proposed systematic review, network meta-analysis and cost-effectiveness analysis will address important questions about the relative costs or risks and benefits of different smoking cessation pharmacotherapies. It is expected that the study findings will be incorporated into future updates of the new NICE Public Health guidance on Smoking Cessation Interventions and Services, a partial update of NICE Public Health guidance on Smoking Cessation Services,[4] expected to be published in November 2017. Critically, we believe that the study findings will offer patients, prescribers and regulators updated and comprehensive information on the safety

and effectiveness of these smoking cessation medicines that will allow them to make informed decisions when evaluating or selecting a pharmacological treatment to assist with a quit attempt.

## ETHICS AND DISSEMINATION

Ethics approval is not required for this evidence synthesis study as it involves analysis of secondary data from RCTs and observational studies. We anticipate dissemination to the following groups, for whom the results of this research will be of interest: the general public, clinicians and healthcare practitioners, academics, policy makers and industry. Findings from the study will be disseminated through conventional academic routes such as peer-reviewed publications and presentations and regional, national and international conferences.

**Correction notice** This paper has been amended since it was published Online First. Owing to a scripting error, some of the publisher names in the references were replaced with 'BMJ Publishing Group'. This only affected the full text version, not the PDF. We have since corrected these errors and the correct publishers have been inserted into the references.

**Contributors** KHT, DC and NJW conceived the idea, planned and designed the study protocol. DC designed the network diagram. KHT, DC and NJW planned the data extraction and statistical analyses. NJW and MS planned the economic evaluation. MRM and DG provided critical insight. KHT and MND wrote the first draft of the protocol, and all authors approved and contributed to the final manuscript.

**Funding** This project presents independent research which was funded by the National Institute for Health Research Health Technology Assessment Programme (project number NIHR HTA 15/58/18).

**Disclaimer** The views and opinions expressed therein are those of the authors and do not necessarily reflect those of the Health Technology Assessment Programme, NIHR, NHS or the Department of Health.

**Competing interests** MRM reports grants from Pfizer, grants from Rusan and non-financial support from GlaxoSmithKline, outside the submitted work. DG had a specified relationship with the MHRA in the past (was a member of the MHRA's Pharmacovigilance Expert Advisory Group and received travel expenses and a small fee for meeting attendance and preparation for meetings).

**Patient consent** Detail has been removed from this case description/these case descriptions to ensure anonymity. The editors and reviewers have seen the detailed information available and are satisfied that the information backs up the case the authors are making.

**Provenance and peer review** Not commissioned; externally peer reviewed.

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
