## [Reviewer comments · BMJ Open]

ARTICLE DETAILS

TITLE (PROVISIONAL)	How do smoking cessation medicines compare with respect to their neuropsychiatric safety? A protocol for a systematic review, network meta-analysis and cost-effectiveness analysis
AUTHORS	Thomas, Kyla; Dalili, Michael; Caldwell, Deborah; Gunnell, DJ; Munafò, Marcus; Stevenson, Matt; Welton, Nicky

VERSION 1 - REVIEW

REVIEWER	Henri-Jean Aubin CESP, Fac. de médecine - Univ. Paris-Sud, Fac. de médecine - UVSQ, INSERM, Université Paris-Saclay, AP-HP, Hôpitaux Universitaires Paris-Sud, 94800, Villejuif, France. Henri-Jean Aubin was member of advisory boards for Pfizer, D&A Pharma, Ethypharm, and Lundbeck, and has received sponsorship to attend scientific meetings, speaker honoraria and consultancy fees from Bioprojet, D&A Pharma, Ethypharm, Lundbeck, Merck-Serono, Novartis, and Pfizer
REVIEW RETURNED	26-Dec-2016

GENERAL COMMENTS	This manuscript describes a protocol for a systematic review, network meta-analysis, and cost-effectiveness analysis of neuropsychiatric safety of smoking cessation drugs. While I do think neuropsychiatric safety of smoking cessation drugs (in particular varenicline and bupropion) has been a critical issue, its importance has somewhat decreased at the end of 2016, since the EAGLES RCT publication in the Lancet(1) (not cited in the manuscript), and the very recent FDA removal of the black box warning regarding possible serious neuropsychiatric side effects with varenicline (December 2016). Nevertheless, the planned network meta-analysis, remains an important work to be done. I should be interested by the publication of the results of this work, while I'm not sure of the interest of publishing the protocol in advance, as it is posted on the PROSPERO site. In any case, I am happy to suggest minor changes in the manuscript: 1. Please update the introduction in accordance to my previous comments (Eagles publication, FDA black box removal)2. Please change the abstract, where the primary safety outcome is said to be serious adverse events, while in the text, p 6, it is completely differently defined as major adverse events that may not be serious.3. Regarding the effectiveness analysis, the authors define their
---

	primary outcome as abstinence for a minimum of 6 months. This outcome isn't in accordance with most RCTs that allows a grace period before calculating prolonged abstinence rates. 4. I have concerns regarding the idea of mixing RCTs and observational studies in the network meta-analysis. I would suggest to plan for 2 separate analysis, one with RCTs only (where many confounders are absent by means of randomization), and another analysis, where observational studies are added. References: 1. Anthenelli RM, Benowitz NL, West R, St Aubin L, McRae T, Lawrence D, et al. (2016): Neuropsychiatric safety and efficacy of varenicline, bupropion, and nicotine patch in smokers with and without psychiatric disorders (EAGLES): a double-blind, randomised, placebo-controlled clinical trial. Lancet. 22:30272-30270.
--	---

REVIEWER	Erin Rogers, DrPH New York University School of Medicine, USA
REVIEW RETURNED	31-Jan-2017

GENERAL COMMENTS	This is a well-written and comprehensive manuscript describing the protocol for a study that will conduct a systematic review and network meta-analysis examining the safety, effectiveness and costs of various tobacco cessation medications. I have minor suggestions to improve clarity of methods:  1. In the Interventions section, the phrase "health technologies" in the first sentence is confusing. I suggest changing to "smoking cessation medications." 2. In the Search Strategy section, authors should add whether there will be any time restrictions on the database search and how non-English language articles will be handled (e.g., translated or reviewed by native speakers). 3. In the Effectiveness analysis section, authors should provide citations for their choice of primary effectiveness outcomes.
--

VERSION 1 – AUTHOR RESPONSE

Reviewer: 1

Nevertheless, the planned network meta-analysis, remains an important work to be done. I should be interested by the publication of the results of this work, while I'm not sure of the interest of publishing the protocol in advance, as it is posted on the PROSPERO site.

While our protocol is posted on the PROSPERO site, this manuscript offers considerably more detail than our PROSPERO record. Additionally, our funding body encourages authors to publish their protocols in open access peer-reviewed journals in addition to the PROSPERO site.

1. While I do think neuropsychiatric safety of smoking cessation drugs (in particular varenicline and bupropion) has been a critical issue, its importance has somewhat decreased at the end of 2016, since the EAGLES RCT publication in the *Lancet*(1) (not cited in the manuscript), and the very recent FDA removal of the black box warning regarding possible serious neuropsychiatric side effects with varenicline (December 2016).Please update the introduction in accordance to my previous comments

(Eagles publication, FDA black box removal).

You are quite right that these recent outputs should be reflected in our manuscript. We submitted our manuscript on December 2nd, prior to confirmation of the warning's removal in the FDA's Drug Safety Communications release on December 16th, and so at that point in time we were unable to include this information in the manuscript. We acknowledge the importance and relevance of these recent developments and have updated the introduction with reference to the EAGLES publication and the FDA's removal of the black box warning on page 3 of the manuscript.

2. Please change the abstract, where the primary safety outcome is said to be serious adverse events, while in the text, p 6, it is completely differently defined as major adverse events that may not be serious.

We have edited this section to improve clarity on page 6 of the manuscript. However, we have not altered the abstract as we believe it accurately describes our primary safety outcome.

3. Regarding the effectiveness analysis, the authors define their primary outcome as abstinence for a minimum of 6 months. This outcome isn't in accordance with most RCTs that allows a grace period before calculating prolonged abstinence rates.

We have chosen sustained smoking cessation, defined as abstinence for a minimum of six months as determined by biochemically validated continuous or prolonged abstinence at the longest reported time point in intention to treat analyses, as our primary effectiveness outcome as this is the strictest definition of abstinence. As we are using data from previous Cochrane reviews of smoking cessation medicines (see [1-4]), we need to be consistent with the definitions used in those reviews. This minimum follow-up period uses the definition of abstinence used in each individual trial (i.e. a grace period following a target quit date, if the trial used that in their reporting). We have noted this and added citations to support our choice of primary effectiveness outcome on page 6 of the manuscript.

4. I have concerns regarding the idea of mixing RCTs and observational studies in the network meta-analysis. I would suggest to plan for 2 separate analysis, one with RCTs only (where many confounders are absent by means of randomization), and another analysis, where observational studies are added.

You are quite right, and this was also our intention. We do intend to conduct separate analyses for the two outcomes. We have clarified this on page 8 of the manuscript.

Reviewer: 2

1. In the Interventions section, the phrase "health technologies" in the first sentence is confusing. I suggest changing to "smoking cessation medications."

We have made this change to the manuscript on page 5.

2. In the Search Strategy section, authors should add whether there will be any time restrictions on the database search and how non-English language articles will be handled (e.g., translated or reviewed by native speakers).

We have added that there will be no time restrictions on the searches for observational studies to page 7 of the manuscript, while on page 7 we noted "To identify studies for the effectiveness network meta-analyses the search strategies from four Cochrane reviews [1-4] (or updated versions where available) will be modified to identify more recent trials for inclusion in the current study in addition to

their previously identified trials.” Non-English language articles will be reviewed by native speakers prior to obtaining a full translation. We have added these details to page 7 of the manuscript.

3. In the Effectiveness analysis section, authors should provide citations for their choice of primary effectiveness outcomes.

Please see previous response to Reviewer 1, item 3.

References

1. Cahill K, Lindson-Hawley N, Thomas KH, Fanshawe TR, Lancaster T. Nicotine receptor partial agonists for smoking cessation. Cochrane Database of Systematic Reviews 2016(5) doi: 10.1002/14651858.CD006103.pub7[published Online First: Epub Date]].
2. Hughes J, R., Stead LF, Hartmann-Boyce J, Cahill K, Lancaster T. Antidepressants for smoking cessation. Cochrane Database of Systematic Reviews 2014(1) doi: 10.1002/14651858.CD000031.pub4[published Online First: Epub Date]].
3. Stead LF, Perera R, Bullen C, et al. Nicotine replacement therapy for smoking cessation. Cochrane database of systematic reviews (Online) 2012;11
4. Hartmann-Boyce J, McRobbie H, Bullen C, Begh R, Stead LF, Hajek P. Electronic cigarettes for smoking cessation. Cochrane Database of Systematic Reviews 2016(9) doi: 10.1002/14651858.CD010216.pub3[published Online First: Epub Date]].

VERSION 2 – REVIEW

REVIEWER	Henri-Jean Aubin CESP, Fac. de médecine - Univ. Paris-Sud, Fac. de médecine - UVSQ, INSERM, Université Paris-Saclay, AP-HP, Hôpitaux Universitaires Paris-Sud, 94800, Villejuif, France. Henri-Jean Aubin was member of advisory boards for Pfizer, D&A Pharma, Ethypharm, and Lundbeck, and has received sponsorship to attend scientific meetings, speaker honoraria and consultancy fees from Bioprojet, D&A Pharma, Ethypharm, Lundbeck, Merck-Serono, Novartis, and Pfizer. He is also member of the American Society of Clinical Psychopharmacology's Alcohol Clinical Trials (ACTIVE) Group, which is supported by Abbvie, Alkermes, Ethypharm, Lilly, Lundbeck, and Pfizer.
REVIEW RETURNED	04-Apr-2017
GENERAL COMMENTS	The authors have correctly addressed my comments and suggestions.